# Interspecific interactions among functionally diverse frugivores and their outcomes for plant reproduction: A new approach based on camera-trap data and tailored null models

Miriam Selwyn[1]*, Pedro J. Garrote[1], Antonio R. Castilla[1], Jose M. Fedriani[2,3]*

**1** Centro de Ecologia Aplicada "Prof. Baeta Neves" CEABN/InBIO, Instituto Superior de Agronomia, Universidade de Lisboa, Lisboa, Portugal, **2** Centro de Investigaciones sobre Desertificación (CIDE-CSIC) Carretera Moncada - Náquera, Moncada, Valencia, Spain, **3** Estación Biológica de Doñana (EBD - C.S.I.C.), Seville, Spain

* mimselwyn@gmail.com (MS); fedriani@csic.es (JMF)

**Data Availability Statement:** All relevant data is within the manuscript and its Supporting Information files.

## Abstract

The study of plant-frugivore interactions is essential to understand the ecology and evolution of many plant communities. However, very little is known about how interactions among frugivores indirectly affect plant reproductive success. In this study, we examined direct interactions among vertebrate frugivores sharing the same fruit resources. Then, we inferred how the revealed direct interspecific interactions could lead to indirect (positive or negative) effects on reproductive success of fleshy fruited plants. To do so, we developed a new analytical approach that combines camera trap data (spatial location, visitor species, date and time, activity) and tailored null models that allowed us to infer spatial-temporal interactions (attraction, avoidance or indifference) between pairs of frugivore species. To illustrate our approach, we chose to study the system composed by the Mediterranean dwarf palm, *Chamaerops humilis*, the Iberian pear tree, *Pyrus bourgaeana*, and their shared functionally diverse assemblages of vertebrate frugivores in a Mediterranean area of SW Spain. We first assessed the extent to which different pairs of frugivore species tend to visit the same or different fruiting individual plants. Then, for pairs of species that used the same individual plants, we evaluated their spatial-temporal relationship. Our first step showed, for instance, that some prey frugivore species (e.g. lagomorphs) tend to avoid those *C. humilis* individuals that were most visited by their predators (e.g. red foxes). Also, the second step revealed temporal attraction between large wild and domestic frugivore ungulates (e.g. red deer, cows) and medium-sized frugivores (e.g. red foxes) suggesting that large mammals could facilitate the *C. humilis* and *P. bourgaeana* exploitation to other smaller frugivores by making fruits more easily accessible. Finally, our results allowed us to identify direct interaction pathways, that revealed how the mutalistic and antagonistic relations between animal associates derived into indirect effects on both plants seed dispersal success. For instance, we found that large-sized seed predators (e.g. ungulates) had a direct positive effect on the likelihood of visits by legitimate seed dispersers (e.g. red foxes) to both fleshy fruited plants. Then, seed predators showed an indirect positive effect on the plants' reproductive success.

**Funding:** This study has been funded by a grant from the Ministerio de Ciencia, Innovación y Universidades, PGC2018-094808-B-I00, to Jose M. Fedriani. Pedro J. Garrote was supported by a fellowship from Fundação para a Ciência e a Tecnologia, SFRH/BD/130527/2017. Antonio R. Castilla was supported by a fellowship from Fundação para a Ciência e a Tecnologia, Fundação para a Ciência e Tecnologia. Miriam Selwyn was supported by a felowship from Consejería de Educación, Junta de Castilla y León. The funders had no role in study design, data collection and analysis, decision to publish, or preparation of the manuscript.

**Competing interests:** The authors have declared that no competing interests exist.

Our new analytical approach provides a widely applicable framework for further studies on multispecies interactions in different systems beyond plant-frugivore interactions, including plant-pollinator interactions, the exploitation of plants by herbivores, and the use of carcasses by vertebrate scavengers.

## Introduction

A major long-established goal in community ecology and evolutionary biology is to understand how interspecific interactions influence population density, distribution, phenotypes, and genotypes [1], crucial to the selection and evolution of life-history traits. During the few last decades, the study of interspecific interactions has experienced an outstanding progress [2] such as, for example, moving from a traditional pair-wise perspective [3–6] to a more realistic and complex multispecific approach, where multiple species interact with each other [2, 7, 8]. However, most of these studies have focused on a single type of interaction at a time, usually studying either mutualistic or antagonistic interactions among species of particular taxonomic groups [9]. Examples of such interspecific interactions that have been most often investigated separately are competitive interactions among vertebrates [e.g., 10, 11] facilitative interactions among plants [e.g., 12, 13] and mutualistic/antagonistic plant-animal interactions [e.g., 8, 14]. Many of these interaction types take place within the same habitats [15–17] and thus, species involved in one interaction type (e.g. predation, competition) can also participate in other interactions (e.g. plant-animal interactions). However, to our knowledge, studies integrating these interaction types are still scarce [but see 18].

New analytical approaches have emerged allowing coping with the methodological challenge that represents studying systems comprising different species that interact among them in variable ways. For instance, multilayer networks [19], spatially explicit agent-based simulation modeling [20] and tailored null models based on resampling techniques [21] represent powerful tools to investigate such complex systems. In many situations, however, a basic challenge remains as to how to monitor subtle and mixed interactions under suboptimal field conditions, especially when target species are nocturnal, secretive, or otherwise elusive animals (e.g. many vertebrates). Importantly, during the last few decades camera traps have revolutionized wildlife research, enabling the collection of precise photographic evidence of rarely seen species, with relatively little cost [22–24]. Camera traps record very accurate data while barely disturbing the photographed animal, operate continually and silently, providing proof of individuals at a spatially restricted spot (e.g. a fruiting plant, a carcass, a water point), their precise time of visit, and their activity (foraging, perching, fighting, etc.) among other relevant data. Hundreds of thousands, if not millions, of photographs of large numbers of vertebrate species are indeed being systematically collected in diverse habitats worldwide [see review 25]. Surprisingly, until very recently camera trap surveys have been seldom used as a quantitative tool to thoroughly measure direct interactions among individuals of different vertebrate species [but see 26–28].

Here we propose a combination of camera-trap survey data and tailored null models as a unifying methodological framework to investigate whether and how interactions between foraging animals (predation, facilitation) alter subsequent plant-animal interactions (pollination, frugivory, herbivory). To illustrate the usefulness of our approach, we evaluate whether interspecific interactions among vertebrate frugivores alter the likelihood of their subsequent interactions with fleshy-fruited plants. Our analytical approach comprises two basic steps. First, we

evaluated if different frugivore species within a study area visited the same fruiting individuals. Then, we used date-time data of photographs and resampling techniques to tailor null models that allowed us to evaluate whether different species of frugivores showed attraction, aversion, or indifference in their timing of visiting (and interacting) with such fruiting plants. Finally, we inferred the indirect consequences of these direct frugivore responses (attractive or aversive) on the plant's reproductive success [e.g., 18].

To illustrate the value of our approach, we chose as study systems the Mediterranean dwarf palm *Chamaerops humilis* and the Iberian pear tree *Pyrus bourgaeana* as well as their shared community of vertebrate frugivores (mostly mammals) at Doñana National Park (SW Spain) [e.g., 18, 29]. This diverse assemblage of frugivores that consume *C. humilis* and *P. bourgaeana*'s fruits can be grouped into five functional frugivore groups: carnivores, wild ungulates, domestic ungulates, pulp feeders (i.e. lagomorphs and rodents) and birds, which may act as seed disperses or seed predators [18, 29–33]; thus, providing a wide spectrum of antagonistic and mutualistic interactions among multiple species across trophic levels [32, 34, 35].

In plant communities, their functional traits vary within environmental gradients and among species occupying similar conditions, raising a challenge for the synthesis of functional and community ecology [36]. The coexistence of consumer species is fostered by resource-use and niche differences, leading to greater resource use in communities with higher number of species [37]. The ecological niche of species is multi-dimensional, including three axes of particular importance that explicitly impact diets and spatial-temporal patterns of abundance: trophic interactions, habitat use and their temporal variability [38]. As a result of interspecific facilitative and competitive interactions, resource partitioning, and niche differentiation, our general hypothesis is that visitation of a fruiting plant by an individual of a given species of vertebrate frugivore could alter (either increasing or lessening) the likelihood of subsequent visits by individuals of other vertebrate frugivore species. Specifically, we expected that (*i*) different frugivores species would tend to visit different fruiting individual plants lessening thus interspecific competition or predation risk, (*ii*) when using the same fruiting plant, prey frugivore species (e.g. small mammals) would tend to temporally avoid plants being foraged by their predators (e.g. carnivores) to reduce predation risk, and (*iii*) large-bodied frugivores (e.g. ungulates) would facilitate the plant exploitation to smaller frugivores by making the fruit more easily accessible (e.g. by giving up ripe fruits in the plants' immediacy).

## Materials and methods

### Study area and sites

The study was carried out during the fruiting seasons (September to December) of 2018 and 2019 in Doñana National Park, SW of the Iberian Peninsula (37° 9′ N, 6° 26′ W, Fig 1a). The region is characterized by sub-humid Mediterranean climate, with hot and dry summers (June-September) and mild, wet winters (November-February [39]). The average annual temperature ranges between 15.4 and 18.7°C (mean ± SE = 16.9 ± 1°C; n = 35; period 1978–2017 and annual precipitation is highly variable, ranging between 170 and 1028 mm (mean ± SE = 542.6 ± 32.8 mm; n = 35; period 1978–2017), with most rainfall during winter (271.4 ± 27.1 mm) and extreme drought during summer (33.3 ± 5.3 mm) (data from Monitoring Team of Natural Process of Doñana Biological Station; http://icts.ebd.csic.es/en/web/icts-ebd/monitoring-program-physical-environment).

We selected two *C. humilis* populations (10 km apart) (Matasgordas and Martinazo sites, Fig 1b) and one population of *P. bourgaeana* located at Matasgordas' site (as density of *P. bourgaeana* is extremely low in Martinazo). Both sites have suffered consequences derived from human activities which have made them differ in vegetation and physiographic characteristics.

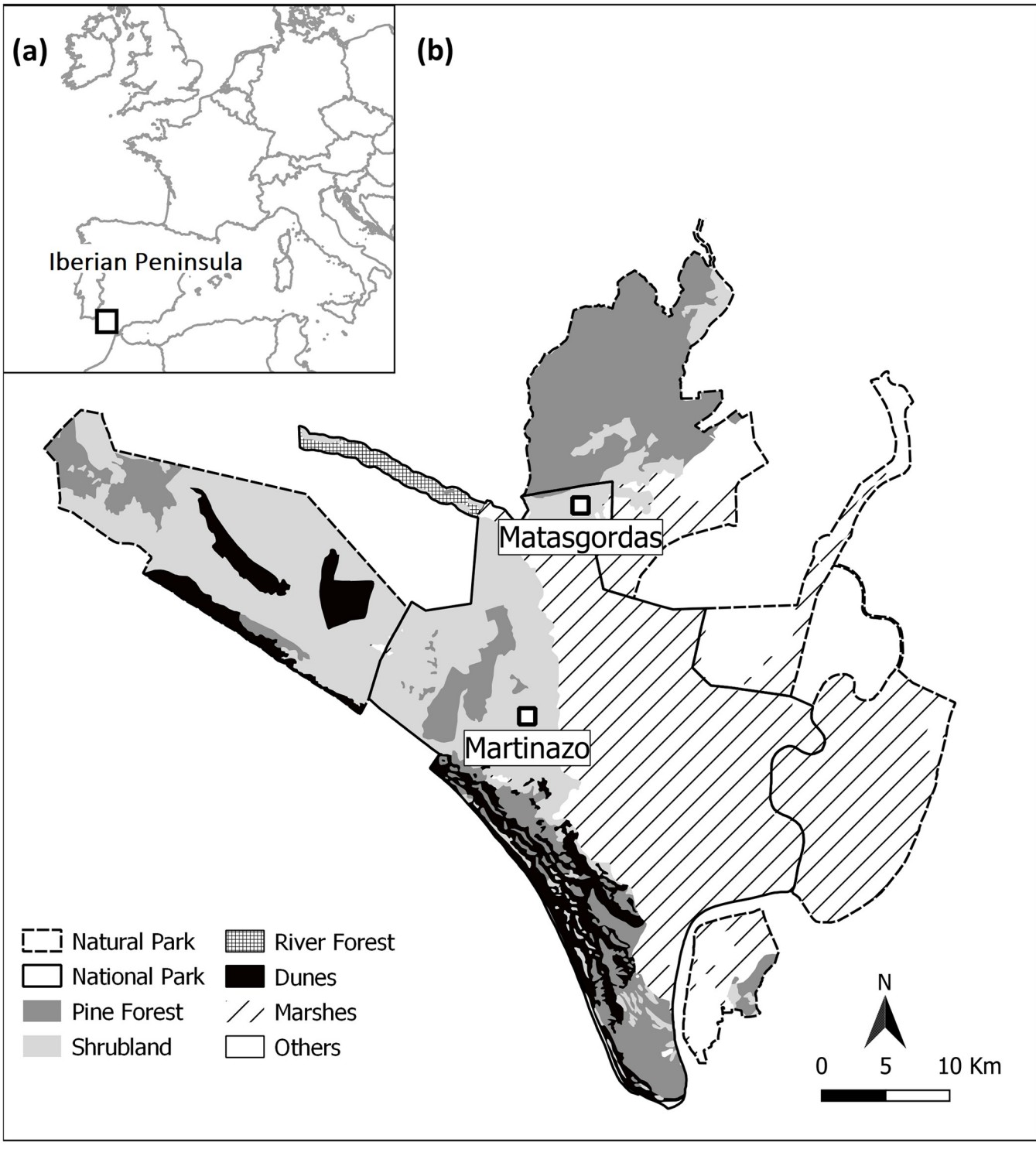

**Fig 1. Study area.** a) Location of Doñana National Park in Europe and the Iberian Peninsula. b) Location of the study sites: Matasgordas (area = 274 ha) and Martinazo (area = 16.7 ha).

In Matasgordas, Mediterranean shrub species (*C. humilis*, *Pistacia lentiscus*, *Halimium halimi-folium*, *Cistus spp.*) as well as some scattered *Pyrus bourgaeana*, *Quercus suber*, *Fraxinus angu-stifolia* and *Olea europaea* var. *sylvestris* trees are slowly recolonizing the old-field area [40–42]. In Martinazo site, the area has been recolonized mainly by early-successional species (*H. halimifolium*, *Ulex spp.*, and *Stauracanthus genistoides*) and animal-dispersed native plants (*C. humilis*, *Rubus ulmifolius*, *Phillyrea angustifolia* or *Asparagus aphyllus* [43]).

## Study species

*Chamaerops humilis* (Mediterranean dwarf palm) and *Pyrus bourgaeana* (Iberian pear tree) are endemic species to the Mediterranean region, specifically distributed in southern Europe and northern Africa [30, 44]. *Chamaerops humilis* is a dioecious palm that blooms from March to May and is mostly pollinated by insects [45–48]. Its fruits are 'polydrupes' (usually 1–3 drupes, which vary from 1–3 cm in diameter [49]) attached to infrutescences of up to 30 cm long (7–120 fruits per infructescence) and usually located at ~10–30 cm above the ground level. The ripening occurs in autumn (September-November). Seedlings emerge during spring and early summer, experiencing high mortality due to summer droughts and herbivory [29]. *Pyrus bourgaeana* is a small (3–6 m height) deciduous monoecious tree which flowers during February and March and is pollinated by numerous Hymenoptera, Diptera, and Coleoptera species [50, 51]. Each tree typically produces 200–450 fruits [29] which are non-dehiscent glo-bose pomes (2–3 cm diameter), with a sugary water-rich pulp, that ripe and drop to the ground from September to December [18]. Seedlings emerge during early spring and experience high mortality due to summer droughts and herbivory [52].

In Doñana National Park, *C. humilis* and *P. bourgaeana*'s seed dispersal is mainly accom-plished by medium-size carnivores such as Eurasian badgers (*Meles meles*) and red foxes (*Vulpes vulpes*) [28, 29]. These two carnivores intensively prey on lagomorphs (*Oryctolagus cuniculus*) and up to five rodent species (mostly *Apodemus sylvaticus* and *Mus spretus*) [32] that feed on *C. humilis* and *P. bourgaeana*'s pulp and seeds [29, 30, 33]. Lagomorphs and rodents act mostly as pulp feeders, although may occasionally act as short-distance seed dis-persers for *C. humilis* [29]. Whereas for *P. bourgaeana*, lagomorphs act mostly as pulp feeders and may occasionally disperse some seeds at short distances, rodents damage all seeds eaten [18, 30]. Fruit predators such as red (*Cervus elaphus*) and fallow deer (*Dama dama*) grind all ingested seeds [29, 30], although can disperse viable seeds of *C. humilis* by regurgitation [31]. Wild boars (*Sus scrofa*) act mostly as seed predators though may occasionally disperse some viable seeds of both species [30, P.J. Garrote *unpublished data*]. Free-ranging cows and horses are abundant in some areas of Doñana and are known to consume and usually predate both *C. humilis* and *P. bourgaeana* seeds. Large mammals (e.g. deer, cows) seem to be better adapted in their foraging, either by breaking many of the defensive needle-like spines in fruiting *C. humilis* or by plucking *P. bourgaeana*'s fruits still attached to the branches (Authors *personal observations*, see S1A File). Hence, ungulates remove many fruits of both plant species and often leave some of them whole or partially consumed around mother plants (Authors *unpub-lished data*). These left fruits are more easily accessible to medium-sized frugivores which hardly cope with the palm's spines and cannot reach the fruits attached to *P. bourgaeana* branches. Therefore, we predict that the foraging of smaller frugivores could be facilitated by larger ones [e.g., 18, 53–55]. Also, birds such as azure-winged magpie *Cyanopica cyanus* and blackcap *Silvia atricapilla* consume *P. bourgaeana*'s fruits mostly acting as pulp-predators rather than seed dispersers [30]. As a result, this diverse assemblage of frugivores can be grouped into the above-mentioned five functional groups (carnivores, wild ungulates, domes-tic ungulates, pulp feeders and birds).

## Data collection

During 2018 fruiting season, we collected data for 29 fruiting *C. humilis* individuals in both study sites (N = 9 for Matasgordas and N = 20 for Martinazo). As for 2019 fruiting season, data was collected for 15 *C. humilis* and 21 *P. bourgaeana* fruiting individuals, all located at Matasgordas site. Data on crop size was recorded for all selected individuals during both study seasons, using visual survey methods (VSM) [56–58]. Camera traps (LTL ACORN 5310A, detection range = 18 m) were installed to collect data regarding to frugivore species visits and use of fruits. Cameras were placed from three to five meters distance from the focal plants and were automatically activated any time a movement occurred, taking a three-photo sequence every second. Camera sampling effort for each individual plant (i.e. camera active days) was carried out during the time period since the fruits were ripe until there were not any fruits left or these were musty. For a given camera and vertebrate species, we considered successive visits separated by more than 5 minutes between them. Although 5 minutes might lead to recording the same individuals several times, we were interested in quantifying successive visits by the same individuals since such accumulated number of visits is likely to alter the behavior of other frugivores. Also, to test our results, we used a stricter criterion, considering independent species visits when the time window was higher than 30 minutes, and therefore individuals were less likely to be resampled [59–61]. For all considered visits, the following data was recorded: date, time, day number since camera was active, frugivore species and their use of the fruiting individual plant. We classified the use of individual plants by each frugivore visitor into two types: (i) recorded mammals and birds that were not physically interacting with the target plant (value = 0) and (ii) mammals or birds clearly interacting with the plant (most likely eating fruit; value = 1) (see S1B and S1C File). Accordingly, we described as total number of visit data all types of plant use (i.e. value = 0 and 1), whilst total number of interaction data refers only to those in which visitors were clearly interacting with the plant (i.e. value = 1). Data on sampling effort days and total number of photographs used in the analysis can be found in S1 Table.

## Analytical approach

To explore our data, we estimated the total number of visits and interactions (as defined above) for each functional group of frugivores (carnivores, wild ungulates, domestic ungulates, pulp feeders and birds) for every focal individual plant. We considered, for each study site, the same pool of vertebrate species due to their high mobility and therefore we did not predict changes in frugivore community structure within each site. Environmental conditions, related to each plant's micro or meso habitat (e.g. crop size), may affect the visitation patterns of different frugivore species [62]. Thus, we assessed the relation between fruit availability (crop size) and total number of visits and interactions by each functional group using Pearson's correlation coefficient applying the function corrplot implemented in the R package corrplot [63].

Because the same frugivore individual could have been photographed several times within a short time period, the analyses described below (i.e. null models) were carried out considering photographs of a given frugivore species at a given individual plant separated by at least 5 minutes (or 30 minutes in a second set of analyses). Our approach was first applied to the whole set of visit data and then to the subset of data corresponding to physical interactions between frugivores and *C. humilis* and *P. bourgaeana* individuals separately.

We assessed the potential interactions between frugivore species by means of three spatially explicit null models that allow coping with the effect of each individual plant's spatial location:

**Null model 1.** We compared the average number of visits by a frugivore species (sp1) to individual plants visited (PV) and not visited (PNV) by a second frugivore species (sp2).

Under the null hypothesis of no interaction between sp1 and sp2, and all else being equal, the average number of visits by sp1 to PV and PNV by sp2 should not differ. If for a given pair of frugivore species the average number of visits by sp1 to PV was significantly larger or smaller than to PNV by sp2 we assumed spatial attraction or avoidance, respectively, between the pair of frugivore species (i.e. sp1 and sp2). For each pair of frugivore species, we examined potential significant differences between the number of visits by one particular species to individual plants visited and not visited by the paired species by fitting Generalized Linear Models with Poisson distribution and log link function [64]. In a second set of analyses, we added "crop size" as a co-variable in the models. Since the new results did not vary regarding the models without the covariate, for major simplicity, we omitted these results.

**Null models 2 and 3.**   To examine temporal interactions among different frugivore species, we compared the *observed mean values of time differences* between successive visits by different frugivore species (OMTD) with the 95% percentile of the *expected time difference* value (95%ETD) based on two different null models [21]. We considered aversive or attractive responses between pairs of frugivore species when OMTD was above or below the 95%ETD limits, respectively. When OMTD was within 95%ETD limits we concluded no significant response between pairs of frugivore species [21].

The metric used to determine observed time differences was the time elapsed (in hours) between the visit of one frugivore species (sp1) until the first visit of a second frugivore species (sp2; i.e. minimum time elapsed between successive interspecific visits). Because we considered the possibility that, for example, sp1 altered the temporal pattern of the plants visits and interactions by sp2 but, for example, sp2 did not alter the temporal pattern of the plants visits and interactions by sp1, the order of species occurrence was taken into account. Thus, for each pair of species, we calculated the time elapsed between a specific species to another (e.g. sp1-sp2) and vice versa (e.g. sp2-sp1). This metric was determined considering the all camera traps with valid data (two camera traps at Martinazo site were discarded) and for all sampling effort days. For all analyses, we only considered visits and interactions of frugivore pairs separated by < 36 hours, as we assume there would not be any species interactions in a greater time window. Interactions between pair of species were only contemplated when the number of observations for OMTD and 95%ETD was ≥4.

For null model 2, the expected minimum time elapsed between successive interspecific visits was calculated for each target pair of frugivore species as for OMTD but considering their occurrences at different and spatially independent (i.e. >100m or >200m apart) plants (S1 Fig). By considering frugivore occurrences at two plants substantially separated we ensured the condition of no potential direct interactions between such frugivore visitors.

For null model 3, calculation of expected time differences between pairs of interspecific frugivore visits was based on the randomization of frugivore occurrence (and timing of occurrence) for each individual plant (S2 Fig). To do so, we run one thousand iterations that assigned occurrence (i.e. visits or interactions) and their timing to a randomly chosen different individual plant. Our procedure preserved not only the observed number of occurrences of each frugivore species, but also their observed circadian rhythms as the observed times linked to each frugivore occurrence (i.e. recorded picture) were not altered. After each iteration, we calculated the time difference between such simulated co-occurrences.

Because results for both *C. humilis* fruiting seasons were consistent, so as not to be redundant, analyses were carried out uniting data of both seasons. Our analytical approach was performed using free software R 3.5.0 [65]. The implemented packages used to tailor the null models were plyr [66], dplyr [67] and lubridate [68]. See S2 File for the main functions used in the analysis.

## Results

### Chamaerops humilis

**Do pairs of frugivore species visit the same fruiting *C. humilis*? (null model 1).** For the complete set of visits in a time window higher than 5 minutes, we found that, in decreasing order, the functional groups with higher number of visits were wild ungulates, domestic ungulates, carnivores, birds and pulp feeders. However, for interaction data the most frequent frugivores were wild ungulates, carnivores, pulp feeders, birds and domestic ungulates (S2 Table).

For the complete visit data, only total number of carnivores' visits was positively correlated with crop size ($r_{5'} = 0.75$, $n_{5'} = 44$, $P_{5'} < 0.001$). Regarding to data corresponding to total number of interactions with *C. humilis* we found that crop size was positively correlated with total number of interactions ($r_{5'} = 0.57$, $n_{5'} = 44$, $P_{5'} < 0.001$), as well as with carnivores ($r_{5'} = 0.78$, $n_{5'} = 44$, $P_{5'} < 0.001$) and wild ungulates ($r_{5'} = 0.44$, $n_{5'} = 44$, $P_{5'} < 0.01$).

In relation to all visits, we found evidence of spatial attraction and segregation in the use of *C. humilis* by some predator-prey species (Fig 2). For instance, on average, badgers visited 2.28 times more often those palms visited by lagomorphs (Fig 2a). Also, lagomorph's visits to palms not visited by red foxes were 8.6 times higher as compared to the mean number of visits to palms visited by red foxes (Fig 2b). Our results showed that palms visited by large ungulates, were more often visited by medium-sized frugivores. For example, red foxes visited palms which were visited by cows 3.34 times more often than those that were not (Fig 2c) and also, visited palms which were visited by horses 7.90 times more often than those that were not (Fig 2d).

**Are there differences between observed and expected minimum time elapsed between successive interspecific visits to *C. humilis*? (null model 2).** When considering the whole set of visits in a time window higher than 5 minutes, null model 2 showed significant interactions between some frugivore pairs. Specifically, the observed mean difference between the time of cow and red fox visits was lower (0.16 times for 100m and 0.14 times for 200 m) than the expected mean difference, therefore indicating strong temporal attraction between both species (Fig 3A and 3B). Also, the observed mean difference between the time of cow and bird was between 0.93 and 0.90 times lower than expected (depending on the criteria used), indicating a moderate attraction between them (Fig 3A and 3B). Conversely, the observed mean differences between the visitation times of horse and badger was 1.94 times greater than the expected mean differences for this frugivore pair, indicating temporal aversion (Fig 3B). As to the subset of data corresponding to interactions between frugivores and *C. humilis*, and separated at least 100 or 200 meters, we found a significant and strong attractive response for the boar-badger pair, which was 0.35 times lower than the expected value for both criterions (Fig 3C and 3D). Also, we found that for a separation of at least 200 meters, that the fox-horse pair showed a significant and attractive response, with the observed mean time difference between the visitations being only 0.62 times the expected value (Fig 3D).

**Are there differences between the observed and the expected time difference between interspecific frugivore visits to *C. humilis*? (null model 3).** When considering all visits within a time window higher than 5 minutes, the difference between the observed and the expected time differences between interspecific visits by frugivore pairs was significant in a few cases. Specifically, the observed mean difference between the pairs cow-red fox and horse-lagomorph visits was much lower (0.17 and 0.30 times, respectively) than the expected mean difference, thus indicating temporal attraction (Fig 4A). Besides, the observed mean differences between the visit times by the lagomorph-badger and lagomorph-rodent pairs was 1.40 and 1.15 times greater, respectively, than the expected mean differences, indicating temporal aversion (Fig 4A). Furthermore, when considering physical interaction data, we found an attractive

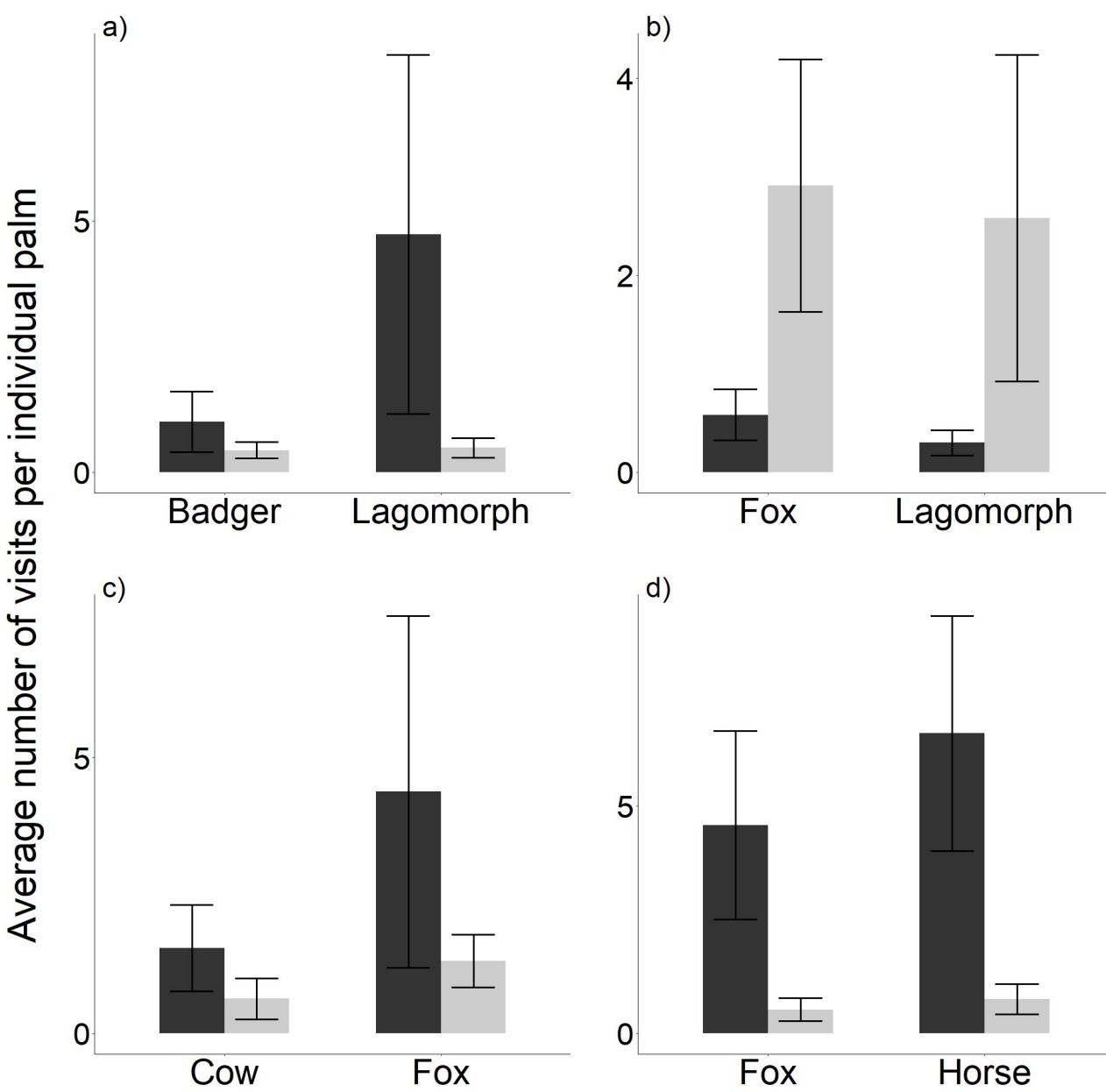

**Fig 2. Difference in *Chamaerops humilis* visit patterns for frugivore pairs of species (null model 1).** The observed values (black bars) represent the mean number of visits by a frugivore species (sp1) to individual plants visited (PV) by a second frugivore species (sp2). The expected values (grey bars) represent the mean number of visits by sp1 to plants not visited (PNV) by sp2. (* *P*<0.05, ** *P*<0.01, *** *P*<0.001).

response for the bird-boar pair, which observed mean difference was 0.33 times lower than the expected one (Fig 4B). Finally, we found a significant aversive response for the lagomorph-rodent pair, which was 1.33 times greater than the expected mean time difference, thus indicating aversion (Fig 4B).

## *Pyrus bourgaeana*

**Do pairs of frugivore species visit the same fruiting *P. bourgaeana*? (null model 1).**  For the complete set of visits in a time window higher than 5 minutes, we found that, in decreasing

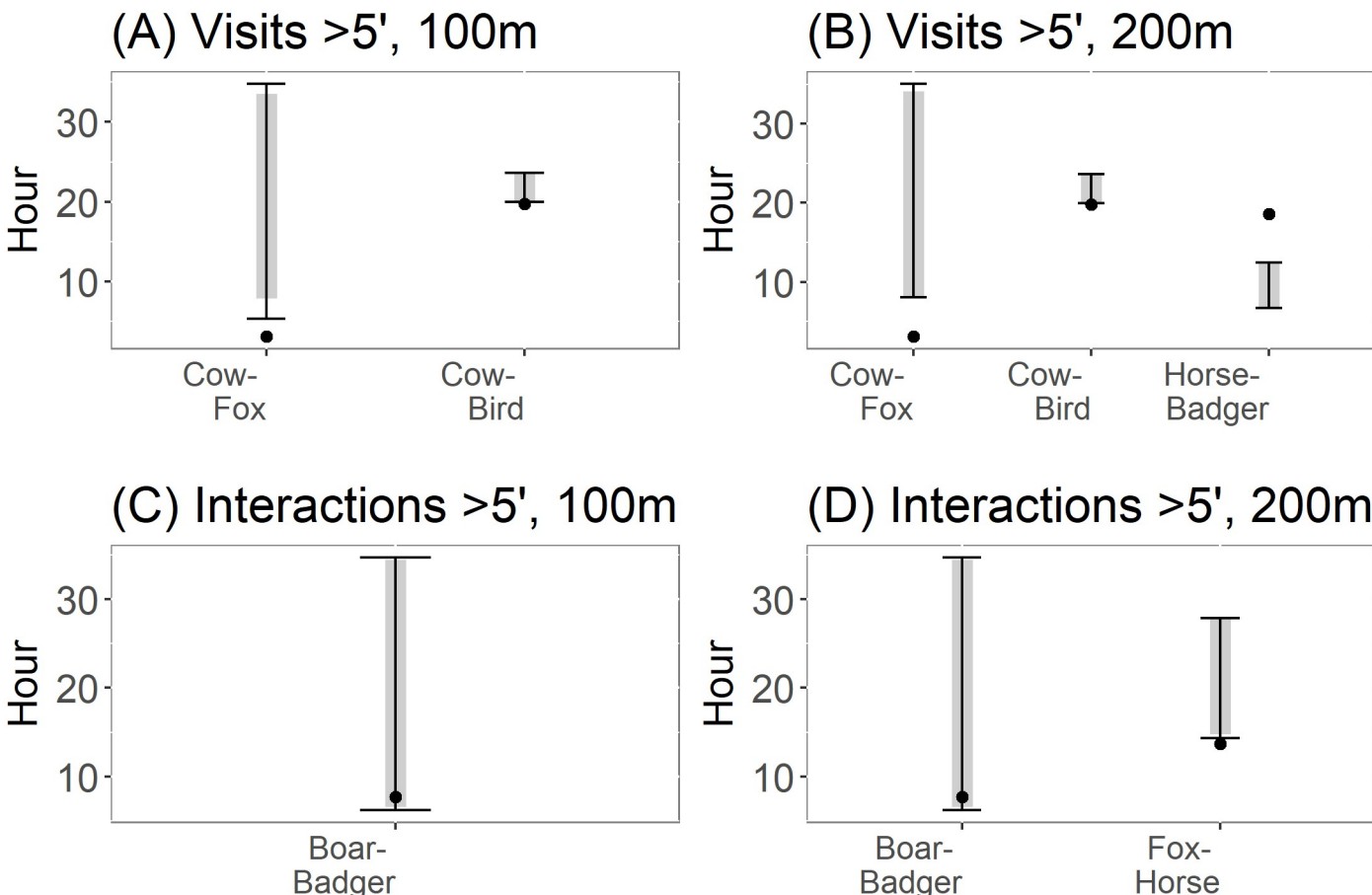

**Fig 3. Significant interactions between frugivores of *Chamaerops humilis* resulting from applying null model 2.** Black dots indicate the observed mean time differences (OMTD). Black lines and grey bars represent the expected 95% and 90% intervals of time elapsed between visits, respectively. (A) Interactions derived from the whole set of visit data, using *C. humilis* individuals at a minimum distance of 100 m. (B) Interactions derived from the whole set of visit data, using *C. humilis* individuals at a minimum distance of 200 m. (C) Interactions derived from data corresponding to frugivores physical interactions with *C. humilis* at a minimum distance of 100 meters. (D) Interactions derived from data corresponding to frugivores physical interactions with *C. humilis* at a minimum distance of 200 meters.

order, the functional frugivore groups with higher number of visits were wild ungulates, pulp feeders, birds and carnivores. However, for interaction data the most frequent frugivores were wild ungulates, pulp feeders, carnivores and birds (S2 Table).

For the complete visit data, total number of carnivores' and wild ungulate visits were positively correlated with crop size ($r_{5'} = 0.54$, $n_{5'} = 21$, $P_{5'} < 0.01$ and $r_{5'} = 0.52$, $n_{5'} = 21$, $P_{5'} < 0.001$, respectively). Regarding to data corresponding to total number of interactions with *P. bourgaeana*, we found that crop size was positively correlated with total number of interactions ($r_{5'} = 0.52$, $n_{5'} = 21$, $P_{5'} < 0.001$), as well as with carnivores ($r_{5'} = 0.55$, $n_{5'} = 21$, $P_{5'} < 0.01$) and wild ungulates ($r_{5'} = 0.52$, $n_{5'} = 21$, $P_{5'} < 0.001$).

In relation to all visits (Fig 5), we found significant evidence of spatial avoidance by lagomorphs which visited 0.98 times less often those *P. bourgaeana* trees visited by badgers (Fig 5a). Also, we observed that trees visited by large ungulates, were more often visited by medium or small-sized frugivores. For example, lagomorphs only visited trees that were visited by boars (Fig 5b) or red deer (Fig 5c).

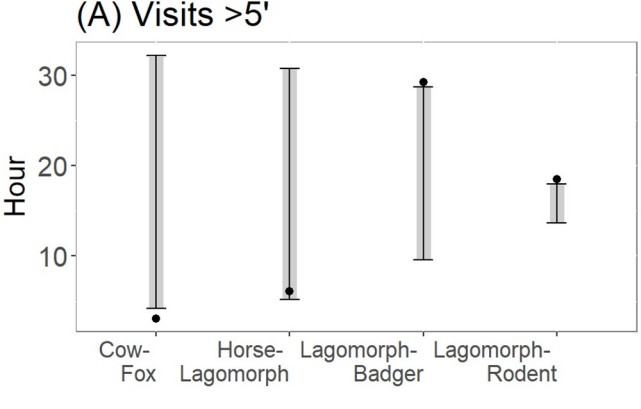

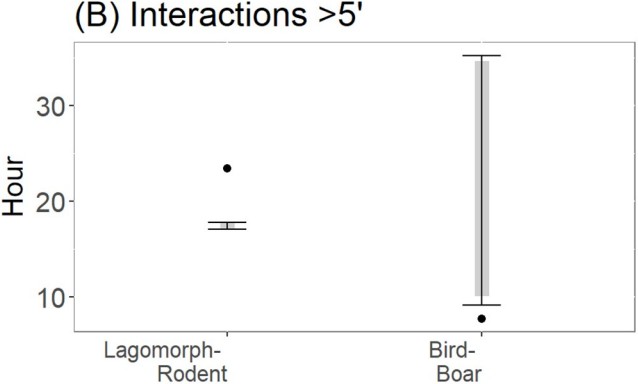

**Fig 4. Significant interactions between frugivores of *Chamaerops humilis* resulting from applying null model 3.** Black dots indicate the observed mean time differences (OMTD). Black lines and grey bars represent the expected 95% and 90% intervals of time elapsed between visits, respectively. (A) Interactions derived from the whole set of visit data. (B) Interactions derived from data corresponding to frugivore physical interactions with *C. humilis*.

**Are there differences between observed and expected minimum time elapsed between successive interspecific visits to *P. bourgaeana*? (null model 2).** When taking into account the whole set of visit data, within a time window higher than 5 minutes, null model 2 showed only one significant response regardless of the spatial independence distance criterion.

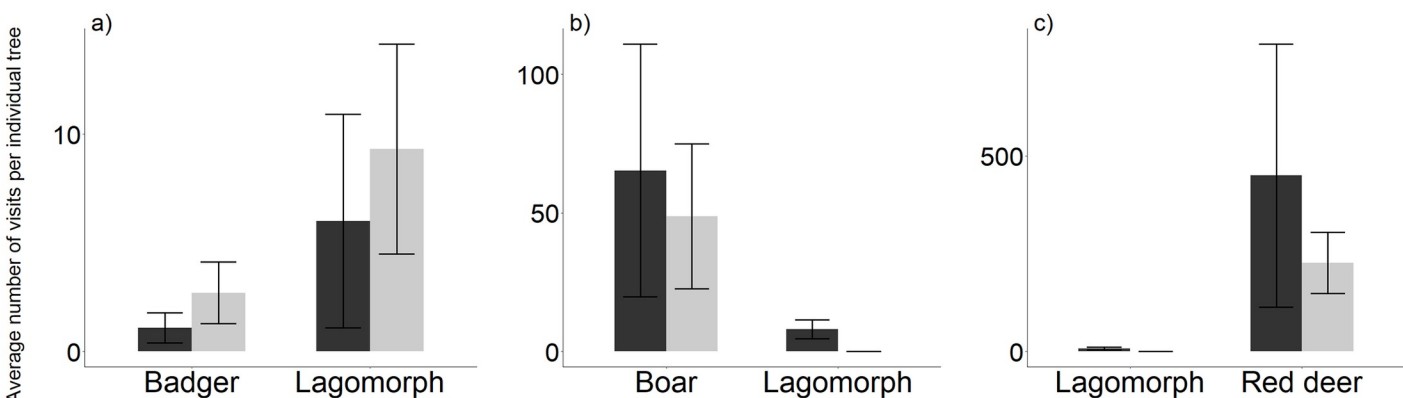

**Fig 5. Difference in *Pyrus bourgaeana* visit patterns for frugivore pairs of species (null model 1).** The observed values (black bars) represent the mean number of visits by a frugivore species (sp1) to individual plants visited (PV) by a second frugivore species (sp2). The expected values (grey bars) represent the mean number of visits by sp1 to plants not visited (PNV) by sp2. (* $P < 0.05$, ** $P < 0.01$, *** $P < 0.001$).

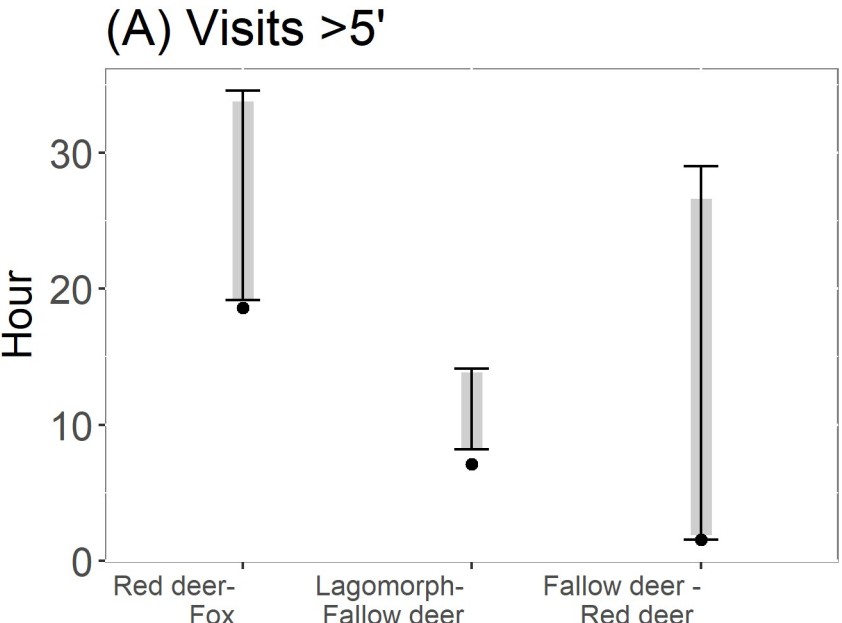

**Fig 6. Significant interactions between frugivores of *Pyrus bourgaeana* resulting from applying null model 3.**
Black dots indicate the observed mean time differences (OMTD). Black lines and grey bars represent the expected 95%
and 90% intervals of time elapsed between visits, respectively. (A) Interactions derived from the whole set of visit data.

Specifically, the observed mean difference between the time of fallow deer and red deer visits
was lower (0.13 times for 100 and 200 meters) than the expected mean difference for this frugi-
vore pair, therefore indicating temporal attraction.

**Are there differences between the observed and the expected time difference between
interspecific frugivore visits to *P. bourgaeana*? (null model 3).** When considering all visits
within a time window higher than 5 minutes, the difference between the observed and the
expected time differences between interspecific visits by frugivore pairs was significant in only
a few cases. Specifically, the observed mean time difference between the pairs red deer-red fox,
lagomorph-fallow deer and fallow deer-red deer visits was always lower (0.73, 0.66 and 0.09
times, respectively) than the expected mean difference, thus indicating temporal attraction
(Fig 6A).

Similar patterns were found when applying the 3 null models for visit and interaction data
within a time window higher than 30 minutes. Thus, regardless of the criteria used to select
the dataset, for most cases, frugivore pairs used the same *C. humilis* and *P. bourgaeana* fruiting
individuals. Most of the resulting interactions were consistent when restricting the analysis to
a narrower time window (30 minutes). Further details on the complete outcome, figures and
statistical summaries (for both 5 and 30 minutes) can be found in S3 File and S3 Table,
respectively.

## Indirect effects on plant reproductive success

As a result of summing up the outcomes of null models 2 and 3, we assessed both direct and
indirect interaction pathways among *C. humilis* and *P. bourgaeana* frugivores' in relation to
these plants reproductive success. For instance, we found that large ungulates such cows or red
deer, had a direct (mostly) negative effect on seed dispersal as they grind most ingested seeds
(Fig 7A). However, cows had a direct positive effect for legitimate seed dispersers, such as red

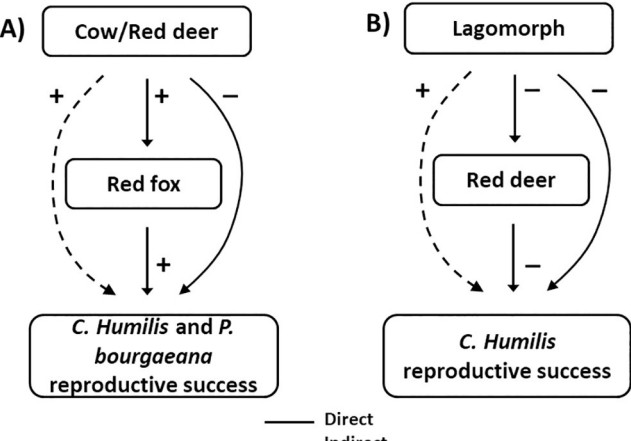

**Fig 7. Effects on *C. humilis* and *P. bourgaeana*'s reproductive success.** Direct and indirect interaction pathways between pairs of frugivore species in relation to *Chamaerops humilis* and *Pyrus bourgaeana*'s reproductive success based on interaction patterns obtained from null models 2 and 3.

foxes which were attracted to palms visited by this large ungulate and thus, lead to an indirect positive effect on *C. humilis*' reproductive success (Fig 7A). Our results also showed a similar situation for the interactions between red deer, red foxes, and *P. bourgaeana* (Fig 7A). Finally, our study also disclosed how, for instance, lagomorphs which mainly had a direct negative effect on *C. humilis* reproductive success acting as pulp feeders, also had a direct negative effect on seed predators such as red deer. Thus, these direct interspecific interactions lead to an indirect positive effect on *C. humilis* reproductive success (Fig 7B).

## Discussion

This study represents one of the first assessments of whether and how functionally diverse vertebrate frugivores alter fruiting plant interactions with subsequent frugivore associates [but see 18]). Specifically, our results revealed, for example, how some frugivore prey (e.g. lagomorphs) avoid those *C. humilis* individuals most often visited by one of its major predators (e.g. red foxes [32]). Additionally, we revealed temporal attraction between large-sized frugivores (e.g. cows, red deer) and medium-sized frugivores (e.g. red foxes) suggesting that ungulates facilitate *C. humilis* and *P. bourgaeana* exploitation to other frugivores by making fruits more accessible. Our analytical approach, based on camera trap data and tailored null models, has also allowed us to identify a number of interactions among frugivores with potential indirect effects on *C. humilis* and *P. bourgaeana*'s reproductive success. Our approach can be widely applied by taking advantage of the large amount of data generated by numerous (on-going and completed) camera-traps surveys done all over the world [25]. This will certainly foster a most comprehensive understanding of the direct and indirect effects of interspecific interactions on ecosystem functioning within and across spatial scales.

### Plant crop size variation and frugivore visitation

Our results revealed that frugivores respond to crop sizes of fruiting plants, by visiting more frequently those plants bearing more fruits. These results are in line with previous studies that have also found a positive relationship between frugivore visitation and plant resource availability [see 69 and references therein]. Interestingly, the positive relationship between crop size and frugivore visitation was mostly driven by two functional groups of frugivores (i.e.

carnivores and wild ungulates). Carnivores (i.e. badgers and foxes) are the most effective seed dispersers of *C. humilis* and *P. bourgaeana* [29, 30], whereas wild ungulates can behave as seed predators or seed dispersers [29–31]. Therefore, a greater dispersal success of individuals bearing more fruits due to higher visitation of the most effective dispersers can be limited depending on whether ungulates behave mostly as dispersers or as seed predators [70]. Other environmental conditions derived from the micro (e.g. plant's location, herbaceous cover, distance to shrub and light availability) and meso habitat (e.g. elevation and type of forest cover) related to each individual plant as well as each animal's niche differences, that allow the coexistence of consumer species, could also exert an effect on the spatial pattern of frugivore visitation [62, 71]. In our system, most of the species (carnivores and ungulates) show high mobility, being able to travel several kilometers within the same day. Therefore, and given the relatively small spatial scale of our study sites, it can be assumed that different species of frugivores had access to all fruiting plants. This scenario however differs for rodents and lagomorphs since they are less mobile; thus, a few cases of spatial avoidance by small mammals identified (null model 1) should be considered with caution.

## Competitive, predatory and facilitative interactions among functionally diverse frugivores

Decrease in fruit display due to exploitation by frugivores is expected to lessen subsequent visitation rates by other fruit consumers [e.g., 72]. In this line, null model 1 results showed that lagomorphs visited less often palms visited by boars, and that red deer visited less often palms visited by rodents, likely because they lessened available ripe fruits. Predator-prey interactions may also affect the spatial and temporal foraging patterns by frugivores [73–75]. Results from null model 1 revealed that lagomorphs tended to avoid *C. humilis* individuals visited by red foxes, likely to reduce predation risk. Also, null models 2 and 3 suggest that badgers seemed to temporally avoid palms visited by lagomorphs. This unexpected result turns less surprising if we consider the fact that badgers do not generally prey upon adult lagomorphs (i.e. the ones that feed on *C. humilis* fruit) but upon small juveniles by digging them out from their burrows [32].

Facilitative interactions in the acquisition of food are highly prevalent among many terrestrial vertebrates and have critical importance in structuring and function of many communities [53–55, 76, 77]. In contrast with Carreira et al.'s [27] findings, we detected that medium-sized frugivores tended to be temporally attracted by larger ungulates. For instance, null models 2 and 3 indicate that red foxes visited those palms previously visited by cows more often than would be expected under the null hypothesis of no interaction between both mammals. This pattern could be a result of ungulates giving up ripe fruits in the palm's immediacy or reducing the number of defensive needle-like spines in fruiting *C. humilis* (Authors *personal observations*) making its fruits more easily accessible to such medium-sized frugivores. Also, null model 3 showed that red foxes seem to be attracted to *P. bourgaeana* individuals previously visited by red deer. This could relate to red deer plucking the fruits from the trees leaving fallen ripe fruits within the tree's immediacy (Authors *personal observations*; see S1C File) which might attract smaller frugivores. Our study shows how this functionally diverse assemblage of frugivores leads to interspecific facilitation in foraging, a widespread pattern in other temperate and tropical habitats [55, 76, 77]. Positive interactions may be an important mechanism linking high diversity to high productivity under stressful environmental conditions [54, 78]. Identifying the specific mechanisms by which frugivore species sense each other is out of the scope of the present investigation, although, signals such odors and trail and territory

marks, amongst others, are frequently perceived by numerous species of terrestrial vertebrates [79–83] and may be involved in our system.

## Community level and indirect effects on *C. humilis* and *P. bourgaeana* reproductive success

Our approach allowed us to identify direct interactions among frugivores of *C. humilis* and *P. bourgaeana* but also potential indirect effects on the reproductive success of plant populations. These indirect effects can either be positive or negative, depending on whether and how they change subsequent visits by antagonistic or mutualistic animal associates. This is an important consideration since the fruits of most plant species are consumed by functionally diverse assemblages of frugivores such as legitimate seed dispersers, seed predators, and pulp feeders [52, 84–86]. As predicted, seed predators and pulp feeders can alter the interaction frequency between fleshy-fruited plants and their mutualistic and antagonistic animal associates [18]. In the first case, ungulate seed predators increased the interaction frequency with legitimate seed dispersers, for both *C. humilis* and *P. bourgaeana* yielding thus an indirect positive effect on seed survival (Fig 7A). Secondly, pulp feeders, that often lessen long-distance dispersal [52], decreased *C. humilis*' foraging by ungulate seed predators, enhancing thus seed survival and fruit removal by mutualistic animal associates (Fig 7B). Therefore, both, mutualistic and antagonistic partners that shape the interaction outcomes among individual plants lead to contrasting seed dispersal success [18, 70], appearing to shape a complex web of direct and indirect effects which's net effect is most likely dependent on the community context [52]. This outcome may be strongly dependent on initial population densities that would most likely alter the dynamics of the system, thus, the net effect of dispersal success, among other factors, will probably be a result of the abundance of different functional guild of frugivores (legitimate seed dispersers, pulp feeders and seed predators) [87–89].

## Conclusions

The combined use of camera trap data and tailored null models has proved to be an effective tool to assess positive and negative interactions between functionally diverse frugivore in relation to a shared resource and infer direct and indirect interaction pathways that influence the reproductive success of fleshy-fruited plants. This new approach may provide advantages and complementarities to ongoing research and methodologies for multispecies interactions such as multilayer networks or spatially explicit agent-based simulation modeling [19–20, 89]. For instance, by relaying on data from camera-trapping, our approach can be applied to uncover interspecific interactions among understudied nocturnal or otherwise elusive animals such as many mammals and other secretive vertebrates. Our analytical approach enabled us to evaluate the specific hypothesis and predictions that we had projected for our particular study system and would hardly have been able to achieve using other similar approaches [see 26–28] that impose a number of requirements and assumptions that did not always fit our specific system.

This study may also foster a greater understanding of the implications of multispecies plant-animal interactions on ecosystem function and restoration by means of direct and indirect interaction pathways that influence seed dispersal and recruitment in disturbed areas [90]. It can also enrich a great potential of interaction network researches predicting indirect novel interactions or changes in functional network structures under global changing conditions (e.g. land use changes, human disturbance and climate change) [35]. Indeed, although we have focused here on interactions among frugivores and two fruiting plants, our approach can be easily expanded at a community level by increasing the sampling effort (i.e. number of camera traps). Finally, it can also be applied to other varied systems and species such as, for example,

plant-herbivore systems [54], plant-pollinator systems [91], or carcass-scavenger systems [92], taking advantage of the large amount of camera-trap surveys that are collecting data all over the world [25].

## Supporting information

**S1 File. Classified frugivore plant use.** A) Red deer plucking the fruits from *Pyrus bourgaeana*. B) Frugivores not interacting with the plant (value = 0). C) Frugivores physically interacting with the plant, most likely eating its fruits (interaction value = 1).
(PDF)

**S2 File. R functions and packages.** Main functions for tailored models 1, 2 and 3 used in the analysis.
(PDF)

**S3 File. Complete outcome, figures and statistical summaries (for both 5 and 30 minutes).** A) Difference in Chamaerops humilis visit patterns for frugivore pairs of species within a time window higher than 5 minutes (null model 1). B) Difference in Chamaerops humilis visit patterns for frugivore pairs of species within a time window higher than 30 minutes (null model 1). C) Zoom of difference in Chamaerops humilis visit patterns for frugivore pairs of species within a time window higher than 5 minutes (null model 1). D) Difference in P. bourgaeana visit patterns for frugivore pairs of species within a time window higher than 5 minutes (null model 1). E) Difference in P. bourgaeana visit patterns for frugivore pairs of species within a time window higher than 30 minutes (null model 1). F) Zoom of difference in P. bourgaeana visit patterns for frugivore pairs of species within a time window higher than 5 minutes (null model 1). G) Significant interactions between frugivores of Chamaerops humilis resulting from applying null model 2 within a time window higher than 30 minutes. H) Significant interactions between frugivores of Chamaerops humilis resulting from applying null model 3 within a time window higher than 30 minutes. I) Significant interactions between frugivores of Pyrus bourgaeana resulting from applying null model 3 within a time window higher than 30 minutes.
(PDF)

**S1 Fig. Null model 2.** Dwarf palms (*Chamaerops humilis*) A and B are spatially independent individuals, in this case, separated by 100 meters from each other (the set of analysis was also run for palm trees separated 200 meters for a stricter spatial independence criterion). For this example, a badger has been recorded at dwarf palm 1 at the time $b_{t0}$. For dwarf palm 2, two species have been recorded twice each: as for the first species, a fox has been recorded at time $f_{t1}$ and time after, another fox has been recorded at time $f_{t2}$. Whilst the second species was a horse recorded at time $h_{t3}$ and after a period of time, another horse has been recorded at time $h_{t4}$. To calculate expected time differences between pairs of interspecific species we have only considered the first successive visit of a different species. Therefore, for this example we have calculated time elapsed between b t0—$h_{t3}$ and $b_{t0}$- $f_{t1}$. This scheme also applies to the Iberian pear (*Pyrus bourgaeana*) tree system.
(TIF)

**S2 Fig. Null model 3.** A, B and C represent *Chamaerops humilis* individuals at which one or both target frugivore groups have been recorded. For this example, a badger has been recorded at dwarf palms A (at time $b_{t1}$) and B (at time $b_{t2}$), and a red fox has been recorded at dwarf palms A (at time $f_{t1}$), B (at time $f_{t2}$) and C (at time $f_{t3}$). To calculate expected time differences between the pair of interspecific species, we randomly assigned occurrence of frugivore groups

(and the timing of the occurrences) by shuffling them 1000 times. When, by chance, both frugivore species (badger and red fox in this case) meet at the same *C. humilis* individual, we calculated time differences between such co-occurrence. In this example we have calculated time elapsed between $b_{t1}$ and $f_{t2}$ for *C. humilis* A and between $b_{t2}$ and $f_{t1}$ for *C. humilis* B. This scheme also applies to the Iberian pear tree (*Pyrus bourgaeana*) system.
(TIF)

**S1 Table. Sampling effort days and total number of photographs used in the analysis for both *C. humilis* and *P. bourgaeana*.**
(PDF)

**S2 Table. Overall patterns.** Total number of visits and interactions for each frugivore functional group and plant species for both 5 and 30 minutes between successive species records.
(PDF)

**S3 Table. Statistical summary for null models 2 and 3.** Table 1) *Chamaerops humilis* 5 minutes statistical summary for null model 2. Table 2) *Chamaerops humilis* 30 minutes statistical summary for null model 2. Table 3) *Chamaerops humilis* 5 minutes statistical summary for null model 3. Table 4) *Chamaerops humilis* 30 minutes statistical summary for null model 3. Table 5) *Pyrus bourgaeana* 5- and 30-minutes statistical summary for null model 2. Table 6) *Pyrus bourgaeana* 5- and 30-minutes statistical summary for null model 3.
(PDF)

## Acknowledgments

We are sincerely grateful to the staff of Doñana National Park for their assistance and to Angelo Santana who kindly helped us with the statistical analyses. Pedro Blendinger, Juliano Bogoni, and Luc Tedonzong provided numerous constructive comments and suggestions that improved a previous version of the manuscript.

## Author Contributions

**Conceptualization:** Miriam Selwyn, Pedro J. Garrote, Antonio R. Castilla, Jose M. Fedriani.

**Data curation:** Miriam Selwyn, Pedro J. Garrote, Antonio R. Castilla, Jose M. Fedriani.

**Formal analysis:** Miriam Selwyn.

**Funding acquisition:** Jose M. Fedriani.

**Investigation:** Pedro J. Garrote, Antonio R. Castilla.

**Project administration:** Pedro J. Garrote, Antonio R. Castilla, Jose M. Fedriani.

**Software:** Miriam Selwyn.

**Supervision:** Pedro J. Garrote, Antonio R. Castilla, Jose M. Fedriani.

**Validation:** Jose M. Fedriani.

**Writing – original draft:** Miriam Selwyn.

**Writing – review & editing:** Pedro J. Garrote, Antonio R. Castilla, Jose M. Fedriani.

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
