## [Decision Letter · Decision Letter 0]

14 Jul 2020

PONE-D-20-17034

Unravelling interspecific interactions among frugivores: a new approach based on camera-trap data and tailored null models

PLOS ONE

Dear Miriam Selwyn

Thank you for submitting your manuscript to PLOS ONE. After careful consideration, we feel that it has merit but does not fully meet PLOS ONE’s publication criteria as it currently stands. Therefore, we invite you to submit a revised version of the manuscript that addresses the points raised during the review process.

Two reviewers have evaluated your manuscript and made constructive contributions. While they found valuable information and were positive, also showed concern on several major issues, so it requires MAJOR CHANGES for your manuscript does meet our criteria for publication. In particular, Reviewer #2 had several major concerns with data analysis.

I mostly agree with the reviewers, and added some comments. You need to solve these shortcomings of the manuscript before it can be considered suitable for publication in this journal. If you can work on the comments, we are looking forward to a new version.

We look forward to receiving your revised manuscript.

Kind regards,

Pedro G. Blendinger, PhD

Academic Editor

PLOS ONE

Journal Requirements:

2. Please note that in order to use the direct billing option the corresponding author must be affiliated with the chosen institute. Please either amend your manuscript or remove this option (via Edit Submission).

Additional Editor Comments:

Consider modifying the title, so that it is not so general and fully reflects the main objectives of the study; see also the comment of Reviewer # 1.Avoid making lack of knowledge a strong point of the study (e.g. lines 53-54, 59-62, 70-71). Instead, highlight why this new knowledge is important, what its contribution is, and how it enables progress on ecological theory.Lines 88-96. Significantly reduce the species description in the Introduction. For the reader unfamiliar with the species on your study site, it is more important to know the functions of the species than their identity. Reviewer's # 1 recommendation on citing evidence of competitive interactions in the system should be included in Methods.Line 132. Please inform on the polydrupe size.Lines 178-191 should be condensed but be careful not to omit relevant information.Line 217. Use italics for “expected time difference”.I do not understand why you show plant crop size variation and frugivore visitation as a separate result. This is confusing and contributes to unnecessarily complexing a manuscript that is already quite complex. Although it is an interesting topic, it is not one of your objectives. In this study, crop size should be more a source of variation to be controlled than a main effect to be evaluated.The potential influence on visit frequency of the fruiting neighborhood does not appear to have been considered. This may be an important limitation of the study; I would expect to see some consideration in this regard in the Discussion. Along the same line, see the comments of both reviewers about the limitations of a non-spatially explicit design in interpreting the results.Results are hard to follow, and even more difficult to interpret without falling into successive ad-hoc explanations. I recommend reducing them, keeping only those most closely related to the main objective and the proposed conceptual framework. In the context of plant-animal interactions in which you framed the study (e.g. lines 76-78), interactions are the most important, not visits. On the other hand, it appears that a 5 min interval between visits may better reflect an effect of responses to previous visitors than a 30 min interval. Therefore, I suggest that you only focus the results in number of interaction using the 5 min as a reference. The other results can be omitted entirely, or can be reduced to the minimum possible in the main text, referring to the supplementary material to see the complete complementary analyzes. This includes the corresponding figures and tables.The Discussion is somewhat poor and does not take advantage of the potential of the results. For example, it could be greatly enriched by focusing more on functional interpretation and the implications for community organization.Figure 8 should move to results.Finally, there is a marked tendency to self-refer, and not always because they are the only references available. It is important that you smooth this trend significantly.

Reviewers' comments:

Reviewer's Responses to Questions

**Comments to the Author**

1. Is the manuscript technically sound, and do the data support the conclusions?

Reviewer #1: Yes

Reviewer #2: Partly

2. Has the statistical analysis been performed appropriately and rigorously? 

Reviewer #1: Yes

Reviewer #2: No

3. Have the authors made all data underlying the findings in their manuscript fully available?

Reviewer #1: Yes

Reviewer #2: Yes

4. Is the manuscript presented in an intelligible fashion and written in standard English?

Reviewer #1: Yes

Reviewer #2: No

5. Review Comments to the Author

Reviewer #1: This manuscript addresses in detail how animal functional groups interact with two fruiting plants, and how some functional groups influence the exploitation of those fruiting plants on a temporal and spatial scale in the Doñana National Park, SW of the Iberian Peninsula. More importantly, the authors provide new a new approach by combining camera trapping and null models to figure out those interactions. It appears that some functional groups of frugivores have negative or positive effects on the exploitation of fruit plants by others and this may have consequences on the reproductive success of fruiting plants. The study opens a new door for studying plant-animal interaction and seed dispersal at the community level to understand the functioning of ecosystems. I was particularly interested in the level of details given in the result section. However, some aspects need clarification in the abstract, introduction, methods, and discussion.

Abstract

Lines 16-20: This sentence refers to lines 73-75. However here the word “combination” that makes the approach new according to me is lacking.

Lines 30-32: This objective seems to be part of the research. So I would suggest considering revising the third sentence of the abstract (lines 15-16) to include this aspect and give here the outcome of this analysis, instead of stating the objective.

Line 35 (carcasses): Maybe "fruit remains", it is generally called "feeding remains"

Introduction

Lines 51-53: Please can the author provide some examples of studies of different types of interactions taking place in the same habitat?

Lines 78-82: a) the authors only treated this in part in the abstract. Please provide information on the outcome of the first step in the abstract. b) Please consider revising the tense of the verbs in both sentences. The first is in the present tense and the second in the past tense.

Lines 82-84: a) The way this research question is stated (using "how") directly means that we already acknowledge that there would be an effect. I would suggest saying, "we inferred the consequence of these direct responses on the plants", instead of "how they would affect".

b) This question seems to form an integral part of the present research, as we can see in the second sentence of the abstract. Although it is mention in the abstract, it is not clear from the title of the manuscript that the interaction between the frugivores and plants will be analyzed.

Line 85: Please consider revising the tense. Some other mistakes may be found elsewhere.

Lines 96-99: The authors have presented the different species or groups of species forming their study system (Lines 85-96). However, there is no clear evidence for the reader why the authors think that the visitation by some species may influence the visitation by other species remain unclear. It would good to set the scene by first talking about "niche partitioning" and "ecological plasticity" that allow ecologically similar species to coexist.

These articles may help:

Salinas‐Ramos, V.B., Ancillotto, L., Bosso, L., Sánchez‐Cordero, V. and Russo, D. (2020), Interspecific competition in bats: state of knowledge and research challenges. Mam Rev, 50: 68-81. doi:10.1111/mam.12180

Ruczyńki L, Zahorowicz P, Borowik T, Hałat Z (2017) Activity patterns of two syntopic and closely related aerial‐hawking bat species during the breeding season in Bialowieza Primaeval Forest. Mammal Research 62: 65– 73.

Schimpp SA, Li H, Kalcounis‐Rueppell MC (2018) Determining species specific nightly bat activity in sites with varying urban intensity. Urban Ecosystems 21: 541– 550.

Line 101: Please clarify in which context the differential visitation of different fruiting plants by different frugivores will contribute to lessening predation risk. To my knowledge, no species was presented in the system as a predator of another in the same system.

Lines 101-103: As mentioned in my previous comment of line 101, it is only here that it appears that some species are the prey of others in this study system. Please make it clear in the description of the study system (lines 85-96). It would also be good to mention some evidence from the literature showing which species in the system are known to compete for food.

Lines 103-105: It is difficult to understand in which ecological context the authors are testing this hypothesis. The authors may provide some evidence or state a theory that will help understand the relationship between the larger and smaller frugivores.

Material and methods

Line 109: Can the author please present a map of the study area so that the reader can quickly have an idea on the scale of the study?

Line 150: In my opinion, this prediction should be in the introduction, and may certainly help understand hypothesis 3, as in the previous comment. Giving evidence from the literature will be important.

Lines 160-163: I would like to know if there any reason why data were not collected for C. humilis at Martinazo in 2019, and why data where not collected for P. bourgaeana in 2018?

Line 163: a) Was there any reference size for each species to be selected in the study sample?

b) Until the end of the document, I have not seen what was the importance of measuring the plant size. Please can you clarify?

Line 205: A part of the aim of this research seems to be an example of an analytical approach for the interactions between animal communities and plants. However, the reason for the choice for the main statistical approach (Null model) has not been made clear to the reader. It would be good to know if there exists another method and why the choice for this particular method.

Lines 206-2012: Please consider revising. The relationship between the sp1 and the sp2 is not very clear.

Line 250: For reproducibility of the method, I encourage the authors to indicate the packages used for the analyses, as well as the principal functions.

Results

Line 254: When talking about "the most abundant visitors", are you referring to the species with the highest number of visits? If so, please consider revising.

Discussion

1) The results showed the influence of some animal species on the visitation rates of some individual plants by other animal species. However, the authors did not discuss the fact that the spatial locations of individual plants coupled with the spatial distribution of the different animal functional groups may influence such interaction; Although all animal species visited all plant individuals. I suggest discussing this aspect and including it as a limit of the study.

I suggest reading these articles:

García, D., Zamora, R. and Amico, G.C. (2011), The spatial scale of plant–animal interactions: effects of resource availability and habitat structure. Ecological Monographs, 81: 103-121. https://doi.org/10.1890/10-0470.1

Morales, J.M. and Vázquez, D.P. (2008), The effect of space in plant–animal mutualistic networks: insights from a simulation study. Oikos, 117: 1362-1370. https://doi.org/10.1111/j.0030-1299.2008.16737.x

2) Secondly, the authors presented their research using a new approach (combining camera traps and null models). However, they did not discuss their approach with other approaches in the literature and present the advantages of their approach to those in the literature.

Line 542: Change "such" into "such as"

Reviewer #2: This study presents a framework to evaluate interspecific interactions (attraction, avoidance or indifference) among homeoterms vertebrates across a meta-region harboring population of a palm (Chamaerops humilis) and a flesh-fruited tree species (Pyrus bourgaeana). This assay was based on camera-trapping during two seasons of plant fruiting contrasting the empirical results with three approaches of null models. I have only a few, but rather fundamental, concerns about this study, as below. Moreover, some parts of the paper are hard to understand or unnecessarily long, for this, I suppose that the paper audience will be better attracted to a leaner version. I hope that my suggestions can contribute to the improvement of the manuscript.

Major comments:

1) Once your approach is spatially explicit at individual-plant-scale, some environmental characteristics can affect your results? For example, micro- and meso-habitat features at each plant buffer may bias the results in terms of species interactions. If these factors were not considered or controlled in your analysis, I recommend acknowledging this issue in a paragraph in the discussion section. If you have some environmental or spatial co-variables, I suggest to include that in the regression models to understand the role of environmental features in the influence on vertebrates co-occurrence patterns.

2) Further, you assumed that all sites have the same pool of vertebrate species? According to your introduction and details in methods, both fruits used by the widest spectrum of consumers, thereby underpinning the trophic structure according to consumer ecological traits (e.g., your functional groups). You can at least discuss this issue because the difference in the local pool of species (real or potential in the future) can cause a rearrangement or bias in the dynamics that you are evaluating.

3) If you had data of camera-trapping during two successive resources seasons, what is the reason to only “infer” about the seed dispersal/frugivory? I suggest using the values on fruit removal as a function of species co-occurrence in each plant individual into a regression model. This can reveal an interesting contrast between trees that were patrolled more frequently by predators. I think that this quantitative approach can provide results to your implicit concern (i.e., how much the vertebrate (in)direct interactions influence the frugivory, and potentially the quantitative impacts on seed dispersal and plant fitness).

4) I recognized that you evaluated the visitation according to crop size via correlation, but I suppose that the crop size should be addressed as a co-variable in your regression models to predict the (co)visits. I’m afraid this invalidates the mechanics or the interpretation of much of this very nuanced analysis because the vertebrates' attraction and indifference can be strongly linked to crop-size (as your correlation reveals).

4) Once both sites have suffered human-induced alterations, do you have a recommendation derived from your findings for the conservation viewpoint? The novelty of your study is showing an effect of the co-occurrence between frugivores attracted in the space by flesh-fruited plants — an issue little addressed in the literature — but you are assuming a mesocosm equal for all site and plant-individual, for example: (i) the local pool of species; (iii) some environmental features; (iv) past human-induced effects (e.g. disturbance); and (iv) individuals crop-size in each temporal replica.

5) I identified a disequilibrium between the sections of the manuscript. The manuscript is long but the discussion section is condensed to a few general paragraphs (6 considering the modest conclusion). Whereas the result section needs a rethink about the size and fashion. I regret that some specific comments about them are lacking, but I recommend rethink the results section because it is too long. Some specific results could be synthesized in the result section but better developed their ecological implications in the discussion.

5) Figures: Although I recognize that there were several paired-interaction to show, I suggest that the figures of the manuscript deserve a reformulation to attract the paper audience. I encourage moving the current version of figures (e.g. Fig 1, Fig 2, Fig 5, and Fig 6) to the supporting information files. In doing so, you can select only the most important interactions to show in a new simplified figure avoiding different scales in the y-axis and providing the same color to each species.

Minor comments

1) L103-105: “and (iii) large-bodied frugivores (e.g. ungulates) would facilitate the plant exploitation to smaller frugivores by making the fruit more easily accessible (e.g. by giving up ripe fruits in the plants’ immediacy).”

I try to understand this presumed facilitation relationship. It seems counter-intuitive because the tendency of large-bodied frugivores ingests the largest portion of fruits. Thus, the interaction expected for the small-bodied frugivores viewpoint is the competition. Do you need to rewrite this premise with more details (as I see in the S1 File).

2) L163-165: “Data on plant size, as well as crop size was recorded for all selected individuals during both study seasons, using sector estimate techniques and if possible total count of fruits.”

I strongly suggest referencing methodological sentences such as this example. Several seminal studies adopted this approach (and other approaches, such as lines 193-195). Further, this sentence apparently is repeated in lines 195-197.

3) L212-214: “We examined potential significant differences between the average number of visits to PV and PNV in each pair of frugivore species, by fitting Generalized Linear Models with Poisson distribution and log link function.”

You need to depict in more detail how these regression models were performed. Further, I do not understand the reason to use Poisson distribution to continuous data (average number of visits). Is the “average” a count data (sum) to justify the use of Poisson distribution?

4) L249-250: “Our analytical approach was performed using free software R 3.5.0 [48].”

Are the functions to perform the null models self-created by you or the analysis was based on a specific R package. Provide more details (but be cohesive) about the data analysis, to this, open space in the manuscript by synthesizing your results.

6. PLOS authors have the option to publish the peer review history of their article (what does this mean?). If published, this will include your full peer review and any attached files.

Reviewer #1: **Yes: **Luc Roscelin Dongmo Tédonzong

Reviewer #2: No

---

## [Author Response · Author response to Decision Letter 0]

23 Aug 2020

Dear Editor,

On behalf of my coauthors, I would like to thank you for the opportunity to revise and resubmit our

manuscript entitled “Interspecific interactions among functionally diverse frugivores and their outcomes for plant reproduction: a new approach based on camera-trap data and tailored null models” (PONE-D-20-17034).

We are also thankful to the Associate Editor Pedro G. Blendinger and the two reviewers for their valuable feedback on an early draft of this manuscript. We found the reviewers’ and the Editor's comments to be very helpful in revising the manuscript and have carefully considered and responded to each suggestion. In the majority of cases we were successful in incorporating the feedback into our revised manuscript. Please, find attached our Responses to Reviewers. 

We hope that this revised version now meets PLOS ONE's requirements and criteria for publication.

Thank you again for your consideration of our revised manuscript.

Sincerely,

Miriam Selwyn

---

## [Decision Letter · Decision Letter 1]

30 Sep 2020

Interspecific interactions among functionally diverse frugivores and their outcomes for plant reproduction: a new approach based on camera-trap data and tailored null models

PONE-D-20-17034R1

Dear Dr. José M. Fedriani

We are pleased to inform you that your manuscript has been judged scientifically suitable for publication and will be formally accepted for publication once it meets all outstanding technical requirements.

Within one week, you will receive an e-mail detailing the required amendments. When these have been addressed, you will receive a formal acceptance letter and your manuscript will be scheduled for publication.

If your institution or institutions have a press office, please notify them about your upcoming paper to help maximize its impact. If they will be preparing press materials, please inform our press team as soon as possible -- no later than 48 hours after receiving the formal acceptance. Your manuscript will remain under strict press embargo until 2 pm Eastern Time on the date of publication. For more information, please contact onepress@plos.org.

Kind regards,

Pedro G. Blendinger, PhD

Academic Editor

PLOS ONE

Reviewers' comments:

Reviewer's Responses to Questions

**Comments to the Author**

1. If the authors have adequately addressed your comments raised in a previous round of review and you feel that this manuscript is now acceptable for publication, you may indicate that here to bypass the “Comments to the Author” section, enter your conflict of interest statement in the “Confidential to Editor” section, and submit your "Accept" recommendation.

Reviewer #1: All comments have been addressed

Reviewer #2: All comments have been addressed

2. Is the manuscript technically sound, and do the data support the conclusions?

Reviewer #1: Yes

Reviewer #2: Yes

3. Has the statistical analysis been performed appropriately and rigorously? 

Reviewer #1: Yes

Reviewer #2: Yes

4. Have the authors made all data underlying the findings in their manuscript fully available?

Reviewer #1: Yes

Reviewer #2: Yes

5. Is the manuscript presented in an intelligible fashion and written in standard English?

Reviewer #1: Yes

Reviewer #2: Yes

6. Review Comments to the Author

Reviewer #1: (No Response)

Reviewer #2: Selwyn and colleagues presented a much-improved version of the paper, embodying the major concerns raised in the first revision. The manuscript now is very better understandable and fashion. Thus, I do not have new concerns or recommendations for this version. I would like to congratulate the authors for the work.

7. PLOS authors have the option to publish the peer review history of their article (what does this mean?). If published, this will include your full peer review and any attached files.

Reviewer #1: **Yes: **Luc Roscelin Dongmo Tedonzong

Reviewer #2: **Yes: **Juliano André Bogoni

---

## [Editor Report · Acceptance letter]

8 Oct 2020

PONE-D-20-17034R1 

Interspecific interactions among functionally diverse frugivores and their outcomes for plant reproduction: a new approach based on camera-trap data and tailored null models 

Dear Dr. Fedriani:

I'm pleased to inform you that your manuscript has been deemed suitable for publication in PLOS ONE. Congratulations! Your manuscript is now with our production department. 

Kind regards, 

on behalf of

Dr. Pedro G. Blendinger 

Academic Editor

PLOS ONE